# Water Privatization and Inequality: Gini Coefficient for Water Resources in Chile

**Juan Correa-Parra** [1,*] **, José Francisco Vergara-Perucich** [1]**and Carlos Aguirre-Nuñez** [2]

1   Centro Producción del Espacio, Universidad de Las Américas, Providencia 7500975, Chile; jvergara@udla.cl
2   Escuela de Construcción, Universidad de Las Américas, Providencia 7500975, Chile; caguirre@udla.cl
*   Correspondence: juan.correa.correa@edu.udla.cl; Tel.: +56-97-373-9644

**Abstract:** This document makes a comprehensive analysis of the inequality of the water market in Chile, measured by the Gini coefficient method. The situation of water rights in Chile is of particular interest because it is a wholly privatized system, where rights are traded in the market and therefore water is presented as a commodity. This privatization of water in Chile occurred as part of the process of neo-liberalization since the 1981 Water Code. The results of this study indicate that both concentration and inequality in the distribution of water rights are very high, which undermines a just social development process and facilitates the economic exploitation of the environment. It proposes a profound revision of the application of a mercantile logic to a scarce essential resource for life such as water and explores the importance of its role as a national good for public use.

**Keywords:** water; Gini coefficient; concentration; privatization; Chile

## 1. Introduction

The World Health Organization and UNICEF indicated that Chile is the Latin-American country with the highest access to safely managed drinking water services [1]. However, water rights are privatized and diverse conflicts are related to this legal framework regarding a natural resource. For instance, an increasing number of slums in the country have limited access to safe water [2]. The privatization scheme has triggered conflicts between water rights owners and local communities, such as the Mapuche-Huilliche case in the south of the territory [3] and with the Atacama communities in the north [4,5]. Furthermore, the privatized water market has fostered the creation of economic groups related to the provision of this resource to households; nevertheless, the quality of the products is not better than the safe water produced by independent firms and scarce regulation concerning these entrepreneurial activities undermines access to better water for consumers [6]. The water market in Chile has exploited the resource at such a level that, nowadays, a social movement on a national scale has organized to contest the privatization of water and to pursue its nationalization as a public good [7]. There is a contradiction between the global data indicating high access to safely managed drinking water services and the conflicts occurring in the territories related to the water market in Chile. In order to illuminate the potential origins of these conflicts, we investigated water inequalities at global and local levels using a statistical approach. The trigger for the privatization of water started in 1975, with the political-economic project of the dictatorship, in relation to implementing a radical free-market economic model privatizing public services usually allocated at the state level. This process is known as neo-liberalization.

As part of the Chilean process of neo-liberalization, in 1981 the Water Code was created, and this scarce natural resource began to be commercialized, generating negotiable property rights in what is known as the water market. Chile is the only country in the world with fully privatized water. Another particularity in Chile is that the owner of the land has no right over the water that may be

present on that piece of land. Such conditions make the Chilean water market a unique study case, now critical due to the climatic crisis. In general, the creation of a water market can be based on the privatization of rights, institutionally controlled business cycles, pricing according to availability, or determining its free disposal as a fundamental human right for subsistence [8]. In the face of the climate emergency, water scarcity is a challenge and the management of the resource must be adapted to the current planetary crisis [9,10]. It is for this reason that it is relevant to critically review those cases in which water resource management instruments have not been adapted to the climate crisis. This occurs in the Chilean case, which takes on greater relevance in the face of resistance from democratic authorities to recognizing the importance of adapting regulation in the water domain to current times. On 7 January 2020, the Senate of the Republic decided to reject the designation of water as a national good for public use [11], despite the fact that Chile faces a severe drought [12] and that the international literature raises the urgency of rethinking models of governance over natural resources [13]. This article presents relevant evidence regarding the urgency of resuming this discussion by exposing how the Chilean case can illustrate the complexity of creating a market through the privatization of water rights for consumption purposes which leads to a high concentration of ownership of a natural resource, generating speculative scenarios with the commercialization of property and significant inequality in water access. The water market creates a compounded scenario that Mehta et al. [14] define as a scarcity policy, where unequal access to water is naturalized, a decision justified by exclusionary property regimes [15], serving as a strategy to divert attention from other problems such as the causes of inequality or poverty [16]. Therefore, studying the water market in Chile provides a view on different aspects of inequality, and not only in access to water.

When a water market was proposed, it is crucial to understand that this is one of the particularities of Chilean reality, wherein in 1981 water rights were privatized and the values and mechanisms necessary in order to exchange water for money established a market that sets its prices by the law of supply and demand. For Boelens [17], in the study of water rights it is critical to identify the affordability of the resource for users, determining how the owner of the water rights exercises his power by controlling a scarce resource elemental for the development of life. Boelens proposes that power relations will define the distribution of water resources, but the discussion also concerns the restructuring of power relations across society. In the specialized international literature, there is consensus that the water market should not operate like that for any other type of tradable good and requires democratic and institutionalized regulations. In some cases, the development of a water market can improve the efficiency in the distribution of water resources [18], but in the case of weak regulatory institutions, it can generate speculative frameworks that end up reducing the population's access to water [18–20]. The conflictive relationship between water owners and communities has been presented in literature from qualitative approaches, [21,22] so this article contributes by offering a nation-wide perspective on how uneven the water market is from a quantitative approach.

Hence, this article seeks to contribute to the discussion of the effects of generating a water market in a situation of water scarcity, that is to say, a scarcity policy; from the review of its distribution and sustainability, it takes the case of water rights in Chile. Specifically, it studies the inequality of water distribution in Chile as measured by the Gini index. To frame the problem, the article presents a general framework for the origin and scope of water management in Chile, from the first regulations to the creation of a market through privatization in 1981. The data for the study and the methods used are then discussed. The source of information analyzed is based on secondary data obtained from the General Water Directorate of the Ministry of Public Works. These data are georeferenced and the inequality of the allocation of this resource for consumption is studied, i.e., for waters that are extracted without reincorporation into supply channels. The method used is a geo-localized Gini index to review the distribution of the resource spatially. The results argue that privatization optimizes the distribution of the water resource. The proposal is to open discussion based on the specialized international literature in order to rethink how to manage water resources in Chile from the institutional and public policy point of view, considering the important drought that the country faces and in view of the

climate emergency. Although Chile presents good indicators of human access to drinking water in urban areas, this study considers total water rights, as the treatment and management of drinking water for human consumption does not reach more than 12%. In this sense, the concentration of rights and their use in extractive areas presents a clear risk for these positive indicators in the future.

*Historical Review of the Formation of the Water Market in Chile*

In 1819, Bernardo O'Higgins, the architect of Chile's independence, drafted a supreme decree to determine how water should be used, based on a metric configuration of the irrigators, clearly establishing how water should be marketed and assigning responsibility for the correct use of the resource by the controllers of water intake. Already by 1857, water resources had been nominated as national public goods. In view of the emergence of drought episodes, in 1872 several regulations sought to generate mechanisms to manage water in the event of alterations in annual rainfall, focusing on rural agricultural productivity. These were the first approaches to regulating water use in Chile, without a market as such but rather with a system of concessions for use for specific productive purposes.

It was in 1927 that a proper water code was drawn up, within the framework of the civil code, with 476 articles covering a large number of the provisions relating to water resource management and enshrining it as a national good for public use, where the State would govern the modes of allocating the use of the resource for activities other than human consumption. This project saw a set of revisions in 1928 and 1930, until the final configuration of the 1936 water code, where concessions for the use of the resource were designated as part of the State's role. The water code took a more definitive and consensual form at the level of parliament in 1951, the year in which the role of the State was specified, establishing that it was the President of the Republic who would approve applications for water use rights [23]. Then, under the government of Eduardo Frei Montalva and in the midst of the search to improve the functioning of agriculture, in 1967 a new water code was created that emphasized the public domain. Under this code, exploitation would be defined according to a rationalized rate by the central planning agencies, thus limiting the volume of maximum use flows [24]. In this first review, it can be seen that Chile historically privileged the understanding that water was a national good for public use whose fair distribution should be supervised by government authority (i.e., by the President of the Republic himself), with a rationality typical of a protagonist State's role in the national economy using a centralized planning approach. With neoliberalism, the paradigm changed completely. The State reduced its participation in the regulation of productive activities, and in matters of public interest markets were created, such as in social security, education, health and water, among others.

The neo-liberalization of public goods in the case of water will took form in the 1981 Water Code. With the framework provided by the Constitution of the Republic promulgated in 1980, a private property regime would be the focus of this new mechanism for regulating water resources. Thus, the management of this resource would be oriented towards the possibilities of its commercialization and acquisition, without State regulation. The State, on the other hand, would play a role in assigning use rights, but without the effective capacity to control such uses, much less the markets for water use transactions between private parties. In this way, water in Chile becomes a commodity [24]. For Humberto Peña [25], the main characteristics of the Chilean water code can be identified as the following aspects:

- Transfer of water management from the State as a public good to the market as a commodity.
- The enshrining of original water rights to generate ownership.
- The work of inspection and conflict resolution among private parties is organized, giving this role top government.
- Strategic planning for water resource management is generated with a focus on its productive role, without sufficient emphasis on the sustainability of its use.

With these transformations, the territory as a whole entered a process of privatization for commercial purposes. For Carl Bauer, the separation of water rights from land rights would be another

problematic factor since water tributaries crossing a land will not have a direct commercial relationship with the landowner. This will account for the overlap of property rights on the same land. The use of water will not have much territorial weight, but will be valuable as an economic resource, so its primary role for the development of life is subjugated to its commercial efficiency. Making water a commodity via the 1981 Water Code also marks the emergence of a water market that in those years followed the recommendations of the Inter-American Development Bank and the World Bank [21]. The main objective argued by the promoters of these reforms was the need to manage water scarcity. These water policies proposed that access and coverage would be achieved in a better way if the until then public management of the water resource passed to private hands, in search of increasing efficiency and improving management, favoring competitiveness in the sector, eliminating barriers for international companies, reducing regulations, and assuring a 10.3% profitability [16–18]. This was the fundamental argument of neoliberalism [26].

To be precise, the perspective of increasing efficiency and improving management of the resource is defined by Panayotakis [27] as "the order of the day", where governments and elites use the idea of scarcity to legitimize the capitalist system and its logic, just like the privatization or financialization of public goods, reserving an exclusive access to these assets to producers only. This privatization generates severe effects on environmental equilibrium and also on social groups in rural and urban areas. Mehta [28] considers these effects as a those of a new type of policy regime, where the practice of resource control ends up by making vulnerable the most deprived communities.

For Bauer [25], the freedom to buy and sell water rights has led to the revaluation of water resources in certain areas. Still, the adverse effects are related to the normative rigidity of the constitutional and legal framework of the Water Codes. This makes it difficult for regulations to be adapted to the diverse uses of water and the nature of water flows, which eventually leads to the aforementioned potential conflicts over water between companies and communities, or the reduction of efficient management of the resource due to inadequate exploitation, despite scarce conditions. Using this short-term logic, water privatization with a focus on productive activities does not consider the common good but instead aspires to economic return on such activities, thus neglecting the ecological role of water resources, since productive processes have other objectives related to income and commercial efficiency.

In 2005, a change to the Water Code was introduced. Although the water market was not disestablished, a monetary tax was generated on the non-exploitation of water resources by those who held the corresponding water rights, seeking to avoid speculation. According to Peña and Jaeger [29], the objective of preventing speculation and monopolistic frameworks was to generate a better balance between the productive role of water and social needs, in addition to adding sustainability criteria to water resource management. However, it was Peña himself, the author of the reform, who later recognized that it was made in an adverse political climate, in which the Constitution facilitates the hindering of progressive reforms, favorable to the conservative political forces that in Chile tend to prioritize the market over the social function of water [24]. In other words, the reforms were limited, without managing to resolve the conflicts related to the sustainability of the water market, preserving the negative characteristics that tend to concentrate resources and access unequally.

Inequality in the distribution of water is a complex scenario. The Water Code establishes a situation in which the owner of the land has no rights over the water flowing on that land. As the community leader Rodrigo Mundaca indicates, in Chile there are no planning regulations associated with the productive capacities of agricultural soil. Therefore, with the person holding water rights on the one hand and agricultural production on the other, communities' access to water may be affected by the consumption industry [1]. Studying the water market in Chile is a means to observe how a non-comprehensive planning apparatus may undermine community access to natural resources.

## 2. Materials and Methods

The data set used for this study comes from the national water rights databases of registered with the General Water Directorate of the Ministry of Public Works of the Government of Chile,

where original rights and new applications for water rights are identified according to Article 122 of the Water Code. The database corresponds to the National Consolidated Rights, dated 20 January 2020. In particular, we have applied the study to consumptive water rights, that is, those assigned so that water is consumed without returning to its original course. Even so, in the presentation of the results, global water rights are presented, against these consumptive rights [30].

However, it should be reported that the original database presented some consistency problems, so a thorough review of the information had to be carried out. In this process of cleaning up the database, inconsistent information had to be removed. In short, from the national consolidated data, out of a total of 131,124 cases, only 97.62% were used (128,015 permits), given that the rest presented problems of various kinds, including typing errors, records without data on the volume of water transferred and units of measurement that cannot be converted into volumes of water, as is the case with the so-called shares or irrigators, which are measures proportional to the flow of a given course, information that is not entirely suitable for the calculation of the Gini Coefficient. The corrected table is attached as a complementary database annex to this publication and has also been sent to the Ministry of Public Works for further correction.

Based on the above, it was decided to work with the 97.62% of the data already processed, on which different spatial concentration analyses were carried out, starting with the spatialization of the water resource capture points (Figure 1). From the data set, several analyses were executed, specifically, (i) distribution according to rights to the resource, (ii) nature of origin, (iii) volume of water extracted and finally (iv) an analysis of how concentrated the water resource is, based on its Gini coefficient, at a national, zonal and regional scale.

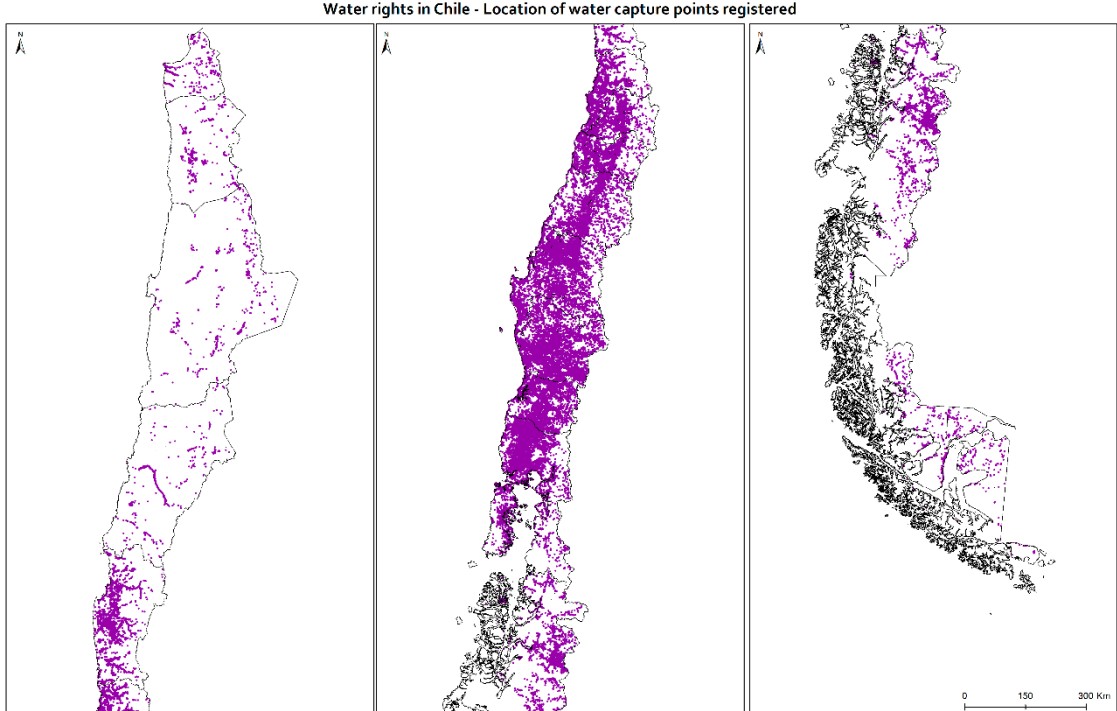

**Figure 1.** Location of water capture points registered as rights at the General Water Directorate of the Ministry of Public Works. Source: drawn up by the authors based on data from the Dirección General de Aguas (DGA).

To analyze this data, a study of unequal access to water rights was carried out using the Gini coefficient. The Gini coefficient is a method for measuring levels of inequality in the distribution of a specific factor in a given population. It is usually used to measure income inequality, as suggested by its creator, Conrado Gini. The result of the calculation ranges from 0 (total equality) to 1

(complete inequality). Its interpretation is simple, which is crucial in a study that seeks to review the problem of water rights in Chile so that its discussion is taken up by different disciplines, from human rights approaches to marketing. In particular, we have classified rights to consumptive water according to the volume of water assigned for each user registered with the General Water Directorate, in order to achieve clarity on the levels of inequality in the allocation of water resources.

Then, based on the distribution patterns of water rights, we will be able to review which actors consume the most greatest amount of liters per second and how these results are interpreted in the light of the water crisis the country is experiencing.

To obtain the Gini coefficient, the following calculation was made:

$$Gini = 1 - \sum_{k=1}^{n-1}(X_{k+1} - X_k)(Y_{k+1} - Y_k) \tag{1}$$

where X corresponds to the cumulative proportion of the variable stakeholders owning water rights included in this study, while Y corresponds to the cumulative proportion of water volume measured in L/s.

## 3. Results

For a general description of the sample, out of a total of 131,124 permits granted and registered in the original database, 128,015 (97.62% of the total) are identified. In this database, 54.1% of the licenses were for groundwater and 45.9% for surface water. Of the total number of permits, those corresponding to consumptive water are studied, whose flow estimate is 4,293,280 L/s. As indicated in Figure 2, most of the non-consumptive licenses are located near the mountain range in the central-southern zone of the country (regions of Valparaíso, Metropolitana, O'Higgins, Maule, Ñuble, Biobio, Araucanía, Los Ríos and Los Lagos), while consumptive licenses are distributed throughout the national territory.

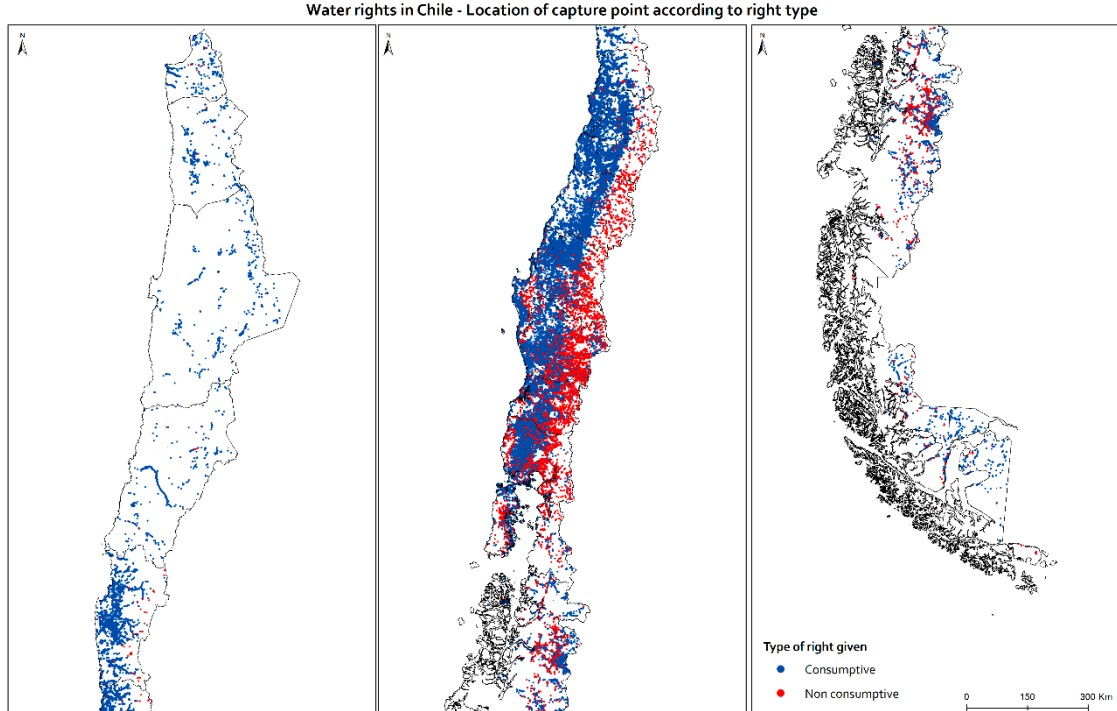

**Figure 2.** National map with the permits according to consumptive or non-consumptive rights. Source: Compiled by the authors based on information from the DGA.

Figure 2 show the distribution of the consumptive and non-consumptive rights throughout the Chilean territory. In addition to the concentration of extraction spots in the central zone of the country

(where about 75% of the population is concentrated), more than the 90% of the non-consumptive rights are localized in this area, particularly in the mountainous area (over 1000 MSL). In these areas, the water rights are linked principally to hydroelectric companies which generate electricity to supply urban areas and for economic activities nearby.

When quantifying the value of the water market for surface consumption rights, it is estimated that it corresponds to a market of USD 45,868,679. One of the problems found in the study is that the General Water Directorate holds 57.4% of permits, without information on the owners or the destination relating to these water rights. For this reason, we present the concentration of water rights, incorporating those protagonists for whom we do not have data (Table 1) and a table of consumptive rights considering only those for which uses for the water are identified (Table 2).

**Table 1.** Consumptive water rights by use, including "No data" cases. Source: Author's elaboration based on DGA data.

| Consumptive Water Rights | | | |
|---|---|---|---|
| **Water Use** | **L/s.** | | **%** |
| | **Recount** | **Add Up** | **Of the Total** |
| No data | 73,591 | 2,465,931 | 57.4% |
| Drink/Domestic Use/Sanitation | 13,008 | 352,034 | 8.2% |
| Hydroelectric Energy | 40 | 5746 | 0.1% |
| Other Uses | 1983 | 95,927 | 2.2% |
| For Observation and Analysis | 3 | 2 | 0.0% |
| Fish Farming | 332 | 21,858 | 0.5% |
| Irrigation | 24,524 | 1,296,526 | 30.2% |
| Silvo-Agropecuario | 127 | 1223 | 0.0% |
| Industrial Use | 380 | 26,915 | 0.6% |
| Medical Use | 4 | 67 | 0.0% |
| Mining use | 479 | 27,050 | 0.6% |
| Total | 114,471 | 4,293,280 | |

**Table 2.** Consumptive water rights by use, excluding "No data" cases. Source: Author's elaboration based on DGA data.

| Consumptive Water Rights | | | |
|---|---|---|---|
| **Water Use** | **L/s.** | | **%** |
| | **Recount** | **Add Up** | **Of the Total** |
| Drink/Domestic Use/Sanitation | 13,008 | 352,034 | 19.3% |
| Hydroelectric Energy | 40 | 5746 | 0.3% |
| Other Uses | 1983 | 95,927 | 5.2% |
| For Observation and Analysis | 3 | 2 | 0.0% |
| Fish Farming | 332 | 21,858 | 1.2% |
| Irrigation | 24,524 | 1,296,526 | 71.0% |
| Silvo-Agropecuario | 127 | 1223 | 0.1% |
| Industrial use | 380 | 26,915 | 1.5% |
| Medical Use | 4 | 67 | 0.0% |
| Mining use | 479 | 27,050 | 1.5% |
| Total | 40,880 | 1,827,348 | |

One of the main observations is that irrigation as a consumptive activity and at the same time a productive activity, registers 71% of the volume of water transferred, with clear identification of use. The use of water for irrigation in Chile is equivalent to the annual consumption of 243 million homes, similar to the number of households in India or 10 times that of the United Kingdom (Figure 3).

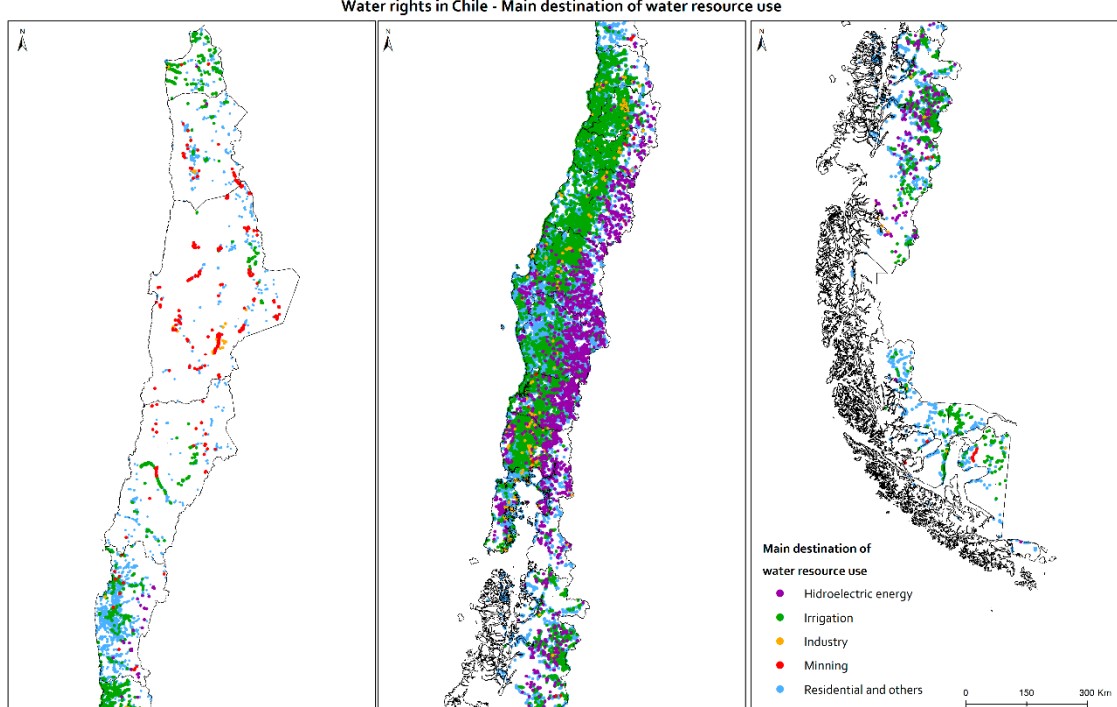

**Figure 3.** Water rights according to the main destination of water use. Source: Author's elaboration according to DGA data.

Figure 3 presents a clear pattern of the distribution of activities using water for production, such as mining activity, which in relation to water rights tenure is predominant in both location and volume, despite the scarcity of water resources in the Atacama Desert, where most mining activities are situated (northern area of the territory). On the other hand, in the central valley of Chile both irrigation and hydroelectric activities capture most of the water rights. This relationship is related to the greater extent of cities, industrial activities, and agriculture. Products with high demand for water are produced in this valley, especially fruit and wines.

This result makes it necessary to review whether this water consumption makes a similar contribution to the treasury through production taxes, when only 2.75% of annual sales are concentrated, representing only 6.27% of companies in Chile and contributing through income tax from agricultural activities only 0.46% of the fiscal budget (0.11% of GDP). It is also essential to review the volume of privatized water distributed in the regions, where Maule (14.69%), O'Higgins (14.04%) and the Metropolitan Region (13.52%) are where a significant part of the total national consumption flow is concentrated (Table 3). This initial descriptive review allows us to recognize the general profile of consumptive water rights in Chile (Figure 4) and the importance of studying levels of inequality and concentration.

**Table 3.** Spatial distribution of permits in the territory by political-administrative region. Source: Compiled by the authors based on DGA data.

| Region | Recount | Add Up | Of the Total |
|---|---|---|---|
| Arica y Parinacota | 2281 | 20,996 | 0.49% |
| Tarapacá | 1708 | 37,273 | 0.87% |
| Antofagasta | 1022 | 25,840 | 0.60% |
| Atacama | 1592 | 41,040 | 0.96% |
| Coquimbo | 11,891 | 149,198 | 3.48% |
| Valparaíso | 13,958 | 410,540 | 9.56% |

**Table 3.** *Cont.*

| Region | Recount | Add Up | Of the Total |
|---|---|---|---|
| Metropolitana | 9575 | 580,271 | 13.52% |
| O'Higgins | 8744 | 602,980 | 14.04% |
| Maule | 9022 | 630,651 | 14.69% |
| Ñuble | 7639 | 161,216 | 3.76% |
| Biobío | 6909 | 333,580 | 7.77% |
| Araucanía | 14,915 | 465,402 | 10.84% |
| De Los Ríos | 8166 | 161,737 | 3.77% |
| De Los Lagos | 10., | 287,634 | 6.70% |
| Aysén | 4704 | 230,327 | 5.36% |
| Magallanes | 1413 | 154,596 | 3.60% |
| Total country | 114,471 | 4,293,280 | |



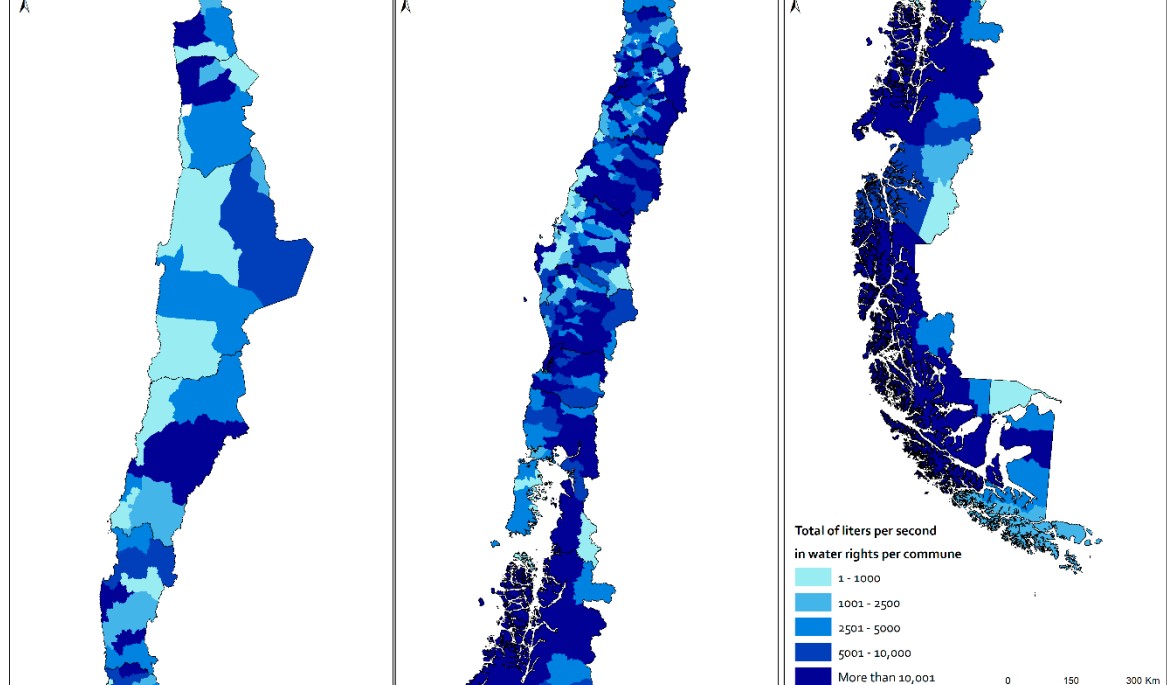

**Figure 4.** Water rights according to total liters/second per commune. Source: Own elaboration from DGA data. Figure 4 shows the distribution and volumes of water rights per commune, which illustrates the territorial inequalities throughout the country. This map underline the high levels of water consumed in the northern areas of the national territory and the high levels of inequality between neighboring communes in the central valley of Chile, where industrial activity, hydroelectric generation, agro-industrial crops and extended cities act as stresses on natural resources. This situation adds up to a critical scenario for the most vulnerable communities, especially small agricultural and livestock producers whose access to water is uneven in relation to the big companies owning water rights [12,13].

The Gini coefficient for all permits at the national level is 0.9999585 and, in the specific case of surface consumption rights, this reaches a value of 0.9537 (Table 4). At first glance, the level of inequality is abysmal, and concentration is very high. In this study, 1% of registered actors own 79.02% of the total volume of water available in the system, which in turn makes up only 4.3% of the existing water property rights. In other words, we see very high inequality and very high concentration of water resources.

**Table 4.** Study of the Gini Coefficient at the national level for the total and specific superficial consumer rights. Source: Prepared by the authors.

| Gini Coefficient for All Rights | |
|---|---|
| Total Rights | 128,015 |
| Total Owners | 63,864 |
| Flow rate (L/s) | 4,865,684,475 |
| Gini Coefficient | 0.999585 |
| **Gini Coefficient for Surface Consumer Rights** | |
| Total Rights | 55,853 |
| Total Owners | 29,001 |
| Flow rate (L/s) | 3,369,691 |
| Gini Coefficient | 0.9537 |

The distribution of water rights in all regions of Chile shows very high inequality (Table 5), fluctuating between 0.8309 (Atacama Region) and 0.9721 (O'Higgins and Ñuble Regions). The most unequal areas correspond to the central sector of the country, where there are greater population and agricultural developments, reflected in regions such as O'Higgins, Maule and Metropolitan, which, although they have fewer water rights, delivered 4726 rights, equivalent to 10.6% of the national total; each right has in proportion more liters/second than the rest of the nation, equal to 42.9% of the national total. In other words, the highest consumption of liters/second occurs in the central valley, in areas with a high preponderance of agricultural activities that demand irrigation, such as vineyards, avocados, berries and fruits in general.

**Table 5.** Summary of Gini Coefficient results by national regions. Source: Prepared by the authors.

| Surface Water Consumption Rights | | | | | |
|---|---|---|---|---|---|
| Region | No Rights | Sum of L/s | National Percentage Rights | National Percentage L/s | Gini Index |
| Arica y Parinacota | 336 | 16,944 | 0.8% | 0.5% | 0.8993 |
| Tarapacá | 285 | 30,596 | 0.6% | 0.9% | 0.9499 |
| Antofagasta | 446 | 11,554 | 1.0% | 0.3% | 0.8980 |
| Atacama | 212 | 11,672 | 0.5% | 0.3% | 0.8309 |
| Coquimbo | 695 | 58,671 | 1.6% | 1.7% | 0.9329 |
| Valparaiso | 2213 | 173,222 | 4.9% | 5.1% | 0.9569 |
| Metropolitan | 1562 | 385,058 | 3.5% | 11.4% | 0.9716 |
| O'Higgins | 1213 | 515,841 | 2.7% | 15.3% | 0.9721 |
| Maule | 1951 | 545,045 | 4.4% | 16.2% | 0.9598 |
| Ñuble | 2312 | 127,003 | 5.2% | 3.8% | 0.9721 |
| Bio Bio | 3888 | 315,563 | 8.7% | 9.4% | 0.9535 |
| Araucanía | 10,121 | 441,378 | 22.6% | 13.1% | 0.9152 |
| De Los Rios | 6482 | 135,056 | 14.5% | 4.0% | 0.8671 |
| De Los Lagos | 7693 | 218,361 | 17.2% | 6.5% | 0.9115 |
| Aysen | 4346 | 229,828 | 9.7% | 6.8% | 0.9704 |
| Magallanes | 974 | 153,899 | 2.2% | 4.6% | 0.8993 |
| Total country | 44,729 | 3,369,692 | 100.0% | 100.0% | 0.954 |

## 4. Discussion

When speculation occurs, decisions are made about specific actions without enough evidence about what is believed to generate optimal results for everyone in the future. A speculative process results from the search for rewards with a strong element of chance. Indeed, speculation has no scientific basis since it is driven by belief rather than evidence. Using this definition as a starting point, we can state that in Chile there is a process of water speculation, given that its consumption would indicate that more water is consumed than is naturally regenerated. A water ownership scheme has

modeled a scenario of scarcity. As Mehta et al. indicate [14], this capitalist fueled scarcity produces a situation in which natural resources have become the focus of global discussion, allowing governments to legitimize the privatized management of resource rights. In the case of Chile, this benefit becomes monetary capital which can be accumulated in rights and used for speculative enterprises which lead to drastic inequality of access to water for people. Restrictions of direct access to user groups benefits a minority with economic and productive interests, principally linked to the social and economic inequalities rooted in Chilean society [31]. Therefore, in our interpretation, water markets reduce accessibility to natural resources because the price of water rights hinders universal access in low-income communities, an everyday reality in rural areas. Given the way in which the water market in Chile is conducted, the use of water rights as financial assets which increase price over time is problematic for social development, and the social benefits of this mode of managing natural resources are difficult to find. Hence, speculation in water rights is one of the results of having a water market for distributing water.

To illustrate this problem with an analogy, it is as if water is being consumed that the glaciers will not run out, while in reality the water supplies of the planet are reaching an abundance threshold, producing scarcity. The mere fact that runoff is much lower than productive consumption generates a deep concern, where sustainability of water resources are compromised in the future. Do we know if the runoff of glaciers is enough for domiciliary consumption and more sustainable forms of production? Some authors [31,32] assert that the naturalization of the scarcity of natural resources, as an acceptable phenomenon not created by society, justifies exclusionary property practices, as with the public right to access nature.

Without regulatory mechanisms appropriate to the Chilean reality [33–35], the water market and its speculative characteristics are dangerous for subsistence in Chile. This complexity, that especially affects rural communities [36], needs to be remedied through public policy and the structural transformation of property management in Chile.

On the one hand, the World Bank proposes that water markets should be formalized and more transparent, which leads to significant price dispersion, unclear operating costs and considerable information asymmetries among market participants. The latter could lead to the development of inequitable economic activities concerning exchange processes. Recently, the Organization for Economic Cooperation and Development (OECD) has proposed that Chile increase its tax collection by increasing (or creating) taxes on the exploitation of natural resources (such as water) to finance new social and infrastructure projects needed to advance development.

The water market is more important in areas where the resource is scarcer [37–39]. However, this is contradicted by much of the literature that presents a critical view of water resource distribution. To understand the idea of the water market, then, we can conclude that water privatization is far from being a socially just way of distributing resources in a society that, in addition, presents high levels of inequality and segregation [40–42].

The climate emergency represents another critical point in the discussion. This crisis will increase the intensity and frequency of extreme events which will cause mass migration waves and food and water insecurity, increasing the occurrence of violent climate conflicts [27,32]. Chile is no exception, considering the social and political conflicts in the Latin-American region, where people from different countries migrate to Chile in search of better opportunities, working in primary economic activities like mining, forestry or agriculture, three activities entirely linked to water rights and community conflicts.

Finally, this work represents the first approximation of an inequality metric and spatial perspective on the concentration of water rights in the continental territory of Chile. With a nation-wide perspective, the article links not only with localization but also indicates the type of rights assigned, the number of liters granted and the ways of using of the resource, all strongly related to the productivity of mining, forestry and agriculture, which are the primary resources of the Chilean economy. We have shared the data freely to contribute to further analysis and studies related to water rights.

**Author Contributions:** Conceptualization, J.F.V.-P.; methodology, J.C.-P.; validation, J.C.-P., C.A.-N. and J.F.V.-P.; formal analysis, J.C.-P., C.A.-N. and J.F.V.-P.; investigation, J.C.-P.; resources, J.C.-P.; data curation, J.C.-P. and J.F.V.-P.; writing—original draft preparation, J.F.V.-P.; writing—review and editing, J.C.-P., C.A.-N. and J.F.V.-P.; visualization, J.C.-P.; funding acquisition, J.F.V.-P. All authors have read and agreed to the published version of the manuscript.

**Funding:** This research was funded by ANID, grant number FONDECYT 11180569 and Universidad de Las Américas.

**Conflicts of Interest:** The authors declare no conflict of interest.

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
