# Peer review of "Water Privatization and Inequality: Gini Coefficient for Water Resources in Chile"

_water, doi:10.3390/w12123369_

Round 1
Reviewer 1 Report
I enjoyed reading this paper and found the methodology interesting and innovative.
It clearly sets out competing uses of water and the inequalities inherent in this. What I found disappointing was the discussion section.
It is very short and doesn't really consider the value of the study in relation to the wider body of literature on inequality in water scarcity e.g. the work by Lyla Mehta.
Developing this section would really enhance the value and impact of the paper.
Author Response
Thanks for this supporting commentary and constructive observations. We agree that the paper was quite focused on the Chilean reality instead of situating its contribution in a broader scenario reaching an international value of the findings. So, as suggested, we developed the discussion section based on international literature. Fundamental were the contributions made by Mehta et al. (2018; 2005), Champeyrache (2014), Panayotakis (2011), among others.
This implied some changes in the introduction, and also in the final section for advancing further research in other scenarios. Please, see the changes in the new version of the article highlighted in yellow.
Reviewer 2 Report
I was excited to read this paper, but more than a little disappointed in the manuscript. It is, as currently presented, more of a research note than a full blown article.
Author Response
We are sorry about the disappointing view. In order to make it more a blown article than a research note, we added different changes to the paper based on new literature incorporated, further discussions and more critical views on the case that can set these findings in a broader research discussion in an international basis.
We counted 150 changes in relation to the original manuscript, so the data and methods are the same but we extended reflections and references to relevant literature. We hope this time the article have a better form. Please, see the changes in the new version of the article highlighted in yellow.
Round 2
Reviewer 1 Report
The revisions give much greater strength to this very interesting paper. I am more than happy to recommend publication.
Reviewer 2 Report
Seems to be improved.
